# Determination of Work Related to Endoscopic Decompression of Lumbar Spinal Stenosis

**DOI:** 10.3390/jpm13040614

**Published:** 2023-03-31

**Authors:** Kai-Uwe Lewandrowski, Morgan P. Lorio

**Affiliations:** 1Center for Advanced Spine Care of Southern Arizona, Tucson, AZ 85712, USA; 2Department of Orthopaedics, Fundación Universitaria Sanitas, Bogotá 111321, Colombia; 3Department of Orthopedics at Hospital Universitário Gaffrée e Guinle, Universidade Federal do Estado do Rio de Janeiro, Rio de Janeiro 20270-004, Brazil; 4Advanced Orthopedics, Altamonte Springs, FL 32701, USA

**Keywords:** relative value units (RVUs), Current Procedural Terminology (CPT^®^), physician payment, CPT^®^ 62380, lumbar herniated disc (LDH), spinal stenosis (LSS), endoscopic decompression

## Abstract

**Background**: Effective 1 January 2017, single-level endoscopic lumbar discectomy received a Category I Current Procedural Terminology (CPT^®^) code 62380. However, no work relative value units (wRVUs) are currently assigned to the procedure. A physician’s payment needs to be updated to commensurate with the work involved in the modern version of the lumbar endoscopic decompression procedure with and without the use of any implants to stabilize the spine. In the United States, the American Medical Association (AMA) and its Specialty Society Relative Value Scale Update Committee (RUC) proposes to the Centers for Medicare and Medicaid Services (CMS) what wRVUs to assign for any endoscopic lumbar surgery codes. **Methods**: The authors conducted an independent survey between May and June 2022 which reached 210 spine surgeons using the TypeForm survey platform. The survey link was sent to them via email and social media. Surgeons were asked to assess the endoscopic procedure’s technical and physical effort, risk, and overall intensity without focusing just on the time required to perform the surgery. Respondents were asked to compare the work involved in modern comprehensive endoscopic spine care with other commonly performed lumbar surgeries. For this purpose, respondents were provided with the verbatim descriptions of 12 other existing comparator CPT^®^ codes and associated wRVUs of common spine surgeries, as well as a typical patient vignette describing an endoscopic lumbar decompression surgery scenario. Respondents were then asked to select the comparator CPT^®^ code most reflective of the technical and physical effort, risk, intensity, and time spent on patient care during the pre-operative, peri- and intra-operative, and post-operative periods of a lumbar endoscopic surgery. Results: Of the 30 spine surgeons who completed the survey, 85.8%, 46.6%, and 14.3% valued the appropriate wRVU for the lumbar endoscopic decompression to be over 13, over 15, and over 20, respectively. Most surgeons (78.5%; <50th percentile) did not think they were adequately compensated. Regarding facility reimbursement, 77.3% of surgeons reported that their healthcare facility struggled to cover the cost with the received compensation. The majority (46.5%) said their facility received less than USD 2000, while another 10.7% reported less than USD 1500 and 17.9% reported less than USD 1000. The professional fee received by surgeons was <USD 1000 for 21.4%, <USD 2000 for 17.9%, and <USD 1500 for 10.7%, resulting in a fee less than USD 2000 for 50% of responding surgeons. Most responding surgeons (92.6%) recommended an endoscopic instrumentation carveout to pay for the added cost of the innovation. **Discussion and Conclusions**: The survey results indicate that most surgeons associate CPT^®^ 62380 with the complexity and intensity of a laminectomy and interbody fusion preparation, considering the work in the epidural space using the contemporary outside-in and interlaminar technique and the work inside the interspace using the inside-out technique. Modern endoscopic spine surgery goes beyond the scope of a simple soft-tissue discectomy. The current iterations of the procedure must be considered to avoid undervaluing its complexity and intensity. Additional undervalued payment scenarios could be created if technological advances continue to replace traditional lumbar spinal fusion protocols with less burdensome, yet no less complex, endoscopic surgeries that necessitate a high surgeon effort in terms of time required to perform the operation and its intensity. These undervalued payment scenarios of physician practices, as well as the facility and malpractice expenses, should be further discussed to arrive at updated CPT^®^ codes reflective of modern comprehensive endoscopic spine care.

## 1. Introduction

Lumbar spinal stenosis is now greatly affecting the aging baby boomer population [1]. Spinal stenosis decompression has been shown to reduce claudication-related disability and improve quality of life [2]. In the elderly, spinal stenosis decompression has become the most common surgical indication [3]. Substantially higher resources are being spent on lumbar spinal stenosis surgery in patients over 65 years of age. In 2007, that cost was estimated to be USD 1.65 billion [4]. In addition, the complexity of the operations performed in these types of patients has increased. One study identified a 15-fold increase in spinal fusion surgery, increasing from 1.3 to 19.9 per 100,000 Medicare beneficiaries from 2002 to 2007 [5]. With this trend also came an increase in life-threatening complications, where the rate increased to 2.3% in decompression patients and 5.6% in complex fusion patients. Similarly, the rehospitalization rate rose to 8% in decompression patients versus 13.0% in complex fusion patients, leading to an increase in adjusted mean hospital charges for complex fusion surgeries of USD 80,888 compared with USD 23,724 for decompression alone [5]. One study illustrated the regional variations in spending related to back surgery. The overall cost increases are staggering. From 1992 to 2003, there has been a 500% increase in money spent on spinal fusion. Expenditures in Medicare patients increased from 75 million dollars to 482 million dollars [6].

Spondylolisthesis [7] and decompression-induced iatrogenic instability [8] have been recognized has indications for lumbar spinal fusion. Endoscopic decompression surgery is increasingly being used as a less burdensome and simplified alternative to more traditional, open, and minimally invasive decompression techniques [9]. While a formal prospective cohort study is currently underway comparing lumbar endoscopic decompression to open decompression and fusion [10], existing studies suggest that lumbar endoscopic stenosis decompression in the central and lateral canal is associated with a low long-term fusion rate, with one study identifying a rate of 2.7% [11] and another, 8.9% [12]. A large body of literature that has been published within the last 5 years shows corroborating results demonstrating its cost-effectiveness [13,14,15,16] and has reported favorable clinical outcomes with endoscopic decompression for lumbar stenosis [17,18]. Many more surgeons are implementing the procedure into their surgical practice portfolio [19]. One study even suggests that endoscopic spine surgery is now the preferred minimally invasive surgery (MIS) performed in the lumbar spine and is more popular than tubular retractor-based microsurgical decompression surgeries [20]. However, implementation hurdles exist, including low reimbursement, high equipment and disposable costs, and the lack of carveouts to cover the cost of the added technology [21].

Effective 1 January 2017, CMS ruled to implement the American Medical Association/Specialty Society (AMA)-approved Current Procedural Terminology (CPT) code 62380 to be used for billing. The code was designated for a single-level endoscopic decompression of the spinal cord and nerve root(s). Its wording included payments for procedural steps, such as laminotomy, partial facetectomy, foraminotomy, discectomy, and/or excision of a herniated intervertebral disc. The original 2017 valuation proposal by the AMA Relative Value Scale Update Committee (RUC) for wRVUs was not implemented. Rather than assigning a final value to CPT code 62380, CMS instead chose to assign contractor pricing, meaning that individual Medicare Administrative Contractors (MACs) could set their own values and make their own reimbursement determination. Considering the technology advances in endoscopic spine surgery in the last five years, CPT^®^ 62380 could remain misvalued as a low-complexity soft-tissue discectomy if and when the RUC is asked to re-consider assigning wRVUs to the code. Many surgeons have begun to perform endoscopic stenosis decompression and spondylolisthesis-related endoscopic fusion surgeries of a much higher complexity and intensity than that associated with an endoscopic discectomy for a symptomatic herniated disc. In an effort to provide the decision makers with new compelling data, an independent survey was conducted with select comparator CPT^®^ codes to estimate the actual workload involved in modern endoscopic lumbar spinal surgeries, including pre-, intra-, and post-operatively while assessing both intensity and complexity of the work.

## 2. Materials and Methods

Endoscopic spine surgeons were surveyed using a previously employed method of estimating the time and complexity of CPT^®^ 62380 by determining comparator CPT^®^ codes and estimated wRVUs [22]. The survey gave a list of reference CPT^®^ codes reflective of common surgeries performed by surgeons within a 90-day global period or representative wRVUs that ranged from below to above 13.5. Table 1 provides these listed CPT^®^ codes used for comparison in this study and their descriptor along with the wRVUs assigned to each by CMS in 2021.

It was assumed that all of the CPT^®^ codes, excluding CPT^®^ 62380, were valued accurately, and only CPT^®^ 62380 was considered to be a potentially misvalued code. Surgeons were reminded that their total work did not just relate to the “skin-to-skin” time spent in surgery (intra-service work). Pre-service work may include completing the patient’s chart and imaging study review, as well as discussing the care plan with the patient and their family or with other doctors. Post-service work is often underestimated as well, such as, for example, the time spent on post-operative care when stabilizing the patient in the recovery room or performing charting work with increasingly more complex and time-consuming requirements related to the documentation and surgeon’s orders dictated by the electronic health record (EMR) system most facilities use nowadays. The increase in surgeon workload related to EMR implementation has essentially gone unnoticed and remains an uncompensated task. Surgeons were advised to take all of these service-related tasks into account and that all work RVUs assigned to the CPT^®^ codes expressed represented a quantitative measure not just of the time and effort involved with delivering the entire service during the 90-day global period, but also a measure of their mental, technical, and physical efforts, the risk involved, and the overall intensity.

The survey was posted on TypeForm (http://www.typeform.com; accessed on 1 May 2022). The authors contacted 210 surgeons to request their participation. The survey was started by 82 endoscopic spine surgeons and completed by 30. It concluded on 15 June 2022. Thus, the completion rate was 36.6%. In the survey app, surgeons could see the comparator CPT^®^ code number, its verbal description by the AMA, and the wRVUs assigned by CMS. As a measure of workload, surgeons were asked to select a CPT^®^ code from the list of comparator codes that best equated to the work involved when performing an endoscopic surgery billed under CPT^®^ code 62380. The surveyed procedure was illustrated in a clinical vignette of a 66-year-old female patient with a 6-month history of progressive leg pain/claudication with some back pain (visual analog scale (VAS) for back pain > 50/100; Oswestry Disability Index (ODI) > 40%) unresponsive to conservative treatments (Figure 1). MRI demonstrated a moderate-to-severe spinal lateral recess and foraminal stenosis at a single level. Flexion/extension radiographs revealed instability (<3 mm) without spondylolisthesis. It is essential to note that the wRVUs of the surveyed procedure continue to be unavailable on the CMS websites, and thus, were not provided as part of the survey.

## 3. Results

The survey results indicate that most surgeons associate CPT^®^ 62380 with the complexity and intensity of a laminectomy and interbody fusion preparation, considering the work in the epidural space using the contemporary outside-in and interlaminar technique and the work inside the interspace using the inside-out technique. Of the responding surgeons, 85.8% valued the wRVU over 13, 46.6% valued it over 15, and 14.3% valued it over 20. Considering multiple responses, surgeons’ revenue for the endoscopic procedure was most frequently generated by commercial insurance (75%), Medicare (70.8%), cash payments (50%), Medicaid (45.8%), personal injury settlements (33.3%), and workmen’s compensation (25%). Only 3 of the 30 surgeons who filled out a complete survey thought they were adequately paid considering the complexity and the intensity of the lumbar endoscopic decompression surgery. Most surgeons (78.5%; <50th percentile) did not think they were adequately compensated. Regarding facility reimbursement, 77.3% of surgeons reported that their healthcare facility struggled to cover the cost with the received compensation (Figure 2).

Regarding facility reimbursement for lumbar endoscopic spine surgery, 25% of responding surgeons did not know what their facility was paid. Four surgeons (14.3%) receiving cash payments indicated that their facility’s payment was under USD 5000. The majority (46.5%) reported their facility receiving less than USD 2000, while another 10.7% reported receiving less than USD 1500 and 17.9% reported receiving less than USD 1000. The professional fee received by surgeons was <USD 1000 for 21.4%, <USD 2000 for 17.9%, and < USD 1500 for 10.7%, meaning the fee was less than USD 2000 for 50% of responding surgeons. Only three surgeons (10%) were paid more than USD 2000 for lumbar endoscopic spine surgery. The majority (89.3%) reported high implementation pricing and disposable costs as problematic at their healthcare facility, making the case (92.6%) for an endoscopic instrumentation carveout to pay for the added cost of technology (Figure 3).

Most responding endoscopic spine surgeons had post-graduate subspecialty training in orthopedic surgery (50%), followed by neurosurgery (46.4%) and pain management (3.6%). Additional carveouts for intra-operative neuromonitoring during awake lumbar endoscopic surgery were recommended by 59.3% of responding surgeons. Regarding payment models, 46.7% of surgeons did not know what arrangement their healthcare facility had with the payors regarding reimbursement of endoscopic spine surgery. Value-based and bundled payment models were reported by 33.3% and 26.7% of surgeons, respectively (Figure 4).

## 4. Discussion

The 2017 Medicare Physician Fee Schedule (MPFS) and the 2017 Hospital Outpatient Prospective Payment System (HOPPS) provide physician and hospital payment rules. The Outpatient Prospective Payment System (OPPS) groups procedures into ambulatory payment classifications (APCs) for outpatient surgery departments at a hospital. In 2022, it listed the Medicare national average CPT 62380 (APC 5114) payment to be USD 6397 [24]. This national average payment number for 2022 was USD 3001 [24]. However, ASC payments can be much lower in certain states, creating an endoscopy implementation hurdle. Considering inflationary cost increases for capital equipment purchases, disposables, payroll, and all other OR-related expenses, the 2017 evaluation of CPT 62380 may need to be higher.

Medical providers submit healthcare claims to Medicare and other health insurance companies using the Healthcare Common Procedure Coding System (HCPCS). This standardized code system provides the framework for consistent and orderly billing. There are HCPCS Level I and HCPCS Level II code sets. The Level I codes are CPT^®^ codes used for billing for medical services provided by physicians, physician extenders, hospitals, laboratories, and outpatient facilities. Medical devices, supplies, medications, transportation services, and other items and services are billed with Level II codes. The resource-based relative value scale (RBRVS) lists the work relative value units (wRVU). A surgeon’s work in endoscopic spine surgery will likely be measured in wRVUs. However, there is no assigned number as of now. Typically, wRVUs are used to assess spine surgeons’ remuneration by considering technical skill, physical and mental effort, judgment, and stress related to patient outcomes. The time needed to perform the surgery is easily measurable. Work RVUs are often used to negotiate surgeons’ salaries in an institutionalized setting without considering the payer mix or the actual collections. In this model, however, surgeons’ pay is typically derived from total work RVUs multiplied by a dollar conversion factor, which in 2022 was USD 34.6062. The conversion factor has been reduced over the last five years, and pandemic-related cost inflation has not been taken into consideration (Table 2).

When attempting to put together a proforma or establish criteria for productivity benchmarking for an endoscopic spinal surgery program, estimating payments for lumbar endoscopic spine surgery is complicated by the lack of an assigned wRVU. The contractor pricing may vary quite a bit between different regions in the United States. The geographic practice cost index (GPCI) adjustments intended to neutralize regional economical variations and which are updated every three years may further decrease or increase payments depending on what area of the United States the endoscopic spine surgery is performed. Our data reflect this dynamic.

What is clear from the authors survey analysis is that nearly half the surgeons thought that their work RVU should be greater than what they are currently reimbursed for simple decompression considering the mental effort, technical effort, physical effort, risk, and overall intensity. The implied reimbursement increase should also include improvements to payments for practice expenses. In our survey, only surgeons receiving cash payments reported to be adequately compensated for endoscopic procedures. Regardless of payer mix, most surgeons (78.5%; <50th percentile) did not think they were adequately compensated. Another 77.3% of surgeons reported that their healthcare facility struggled to cover the cost with the received compensation (Figure 2). This team of authors could only find a payment amount of USD 2803.36 in the medical fee schedule of the Office of Workers’ Compensation Programs (OWCP), effective 20 June 2020, but no RVU numbers were found for work, practice expense (PE), or malpractice (MP) RVUs [26]. This is a marked increase from 2019 where USD 686.08 was advised for payment [27]. The payments for outpatient services at a hospital are not publicly available. No RVUs or dollar amounts were listed in 2021. However, our survey suggests that the actual payments for professional and ASC facility payments are much lower, with the majority (46.5%) reporting their facility receiving less than USD 2000, while 10.7% reported less than USD 1500 and 17.9% reported less than USD 1000. The professional fee received by surgeons was <USD 1000 for 21.4%, <USD 2000 for 17.9%, and <USD 1500 for 10.7%, meaning the fee was less than USD 2000 for 50% of responding surgeons.

Currently, CPT^®^ code 62380 has contractor pricing and no specific RVU numbers. This survey provides the basis for discussions with decision makers in public healthcare systems across the world, including the American Medical Association/Specialty Society Relative Value Scale Update Committee (RUC), regarding the workload involved in endoscopic decompression surgery of the lumbar spine. For example, one could argue that the time spent on low-intensity activities related to placing guidewires and dilators and establishing endoscopic working channel access to the spine is much less and that the majority of the surgery time is spent in high-intensity activities where high concentration is required nearly the entire time, causing increased surgeon stress. In open laminectomy surgery, however, a proportionally higher time of lower intensity is spent on surgical exposure, wound closure than on actual neural element decompression. The increased workload should also account for the steeper learning curve with the endoscopic procedure and the need for more intense skill-based and one-on-one training. Most post-graduate training programs do not offer formal training in endoscopic spine surgery. Currently, most spine surgeons have to be trained on their own time after graduation and incur direct and indirect costs such as course fees, travel expenses, and lost income from missed work and other compensated tasks. Additional cost factors regarding capital equipment purchases, disposables, and practice- and malpractice expenses on the physician and facility side also have to be considered.

Our study highlights the need for more sophisticated and detailed discussions with payors to facilitate the transition from costly traditional open spinal surgery protocols to targeted minimally invasive ones that employ personalized medicine and staged management concepts. The burden of proof is with the innovators, and thus the authors conducted this survey study to obtain a more accurate snapshot of contemporary practice patterns and how the modern endoscopic surgery platform is currently being deployed in the field. Our survey corroborates the previously reported observation that many more surgeons are performing more advanced bony decompression and reconstructive fusion techniques. Additional CPT codes should be recommended for these more complex endoscopic surgeries to adequately compensate surgeons for the increased workload and motivate them to modernize their practice. Spine-related instrumentation or implant carveouts were a rarity in the healthcare facilities where our responding spine surgeons worked. Therefore, regardless of their training background, most surgeon respondents recommended instrumentation carveouts since their facilities struggled to cover the added cost of the capital equipment and disposable purchases needed to implement the innovative endoscopic spine surgery program. Most spine surgeons also suggested additional carveouts for neuromonitoring since many perform the procedure on patients who are awake with sedation under local anesthesia.

## 5. Conclusions

This survey among endoscopic spine surgeons raises awareness that no RVU values have been assigned by the RUC to the CPT^®^ code 62380 since its approval in 2016. Moreover, by analyzing the responses of surgeons regarding appropriate comparator CPT^®^ codes, this survey measures the technical and physical effort, risk, and overall intensity of the endoscopic procedure and not just the time required to perform the surgery. Modern contemporary lumbar endoscopic surgery techniques go way beyond the scope of simple discectomy work for which the CPT^®^ 62380 code was originally intended. Many surgeons perform complex stenosis decompression in the epidural space and the intervertebral disc spaces, and even fusion surgeries with the endoscopic platform. This observation is corroborated by the fact that most surgeons associate CPT^®^ 62380 with the complexity and intensity of a laminectomy and interbody fusion preparation, considering that the work involves using the contemporary outside-in transforaminal and the translaminar interlaminar technique, as well as the inside-out technique to accomplish work in the intervertebral space. Additional assessments are needed to measure the physician and facility practice- and malpractice expenses. Other CPT^®^ codes should be considered for these more advanced procedures. Carveouts are likely required to support the technology rollout and pay for the added cost of the endoscopic surgery program. Additional undervalued payment scenarios could be created if technological advances continue to replace traditional lumbar spinal fusion protocols with less burdensome, yet no less complex, endoscopic surgeries with a high surgeon effort - not just in terms of time required to perform the operation, but in its intensity. These undervalued payment scenarios and physician practice, facility, and malpractice expenses should be further discussed with decision makers in public healthcare systems, including CMS and the Office of Management and Budget (OMB), to arrive at updated payment schedules reflective of modern comprehensive endoscopic spine care. Current RUC methodology does not provide the tools necessary to properly assess highly intense procedures incorporating emerging technology with enhanced value propositions. Parity of endoscopic spine procedures with open spine procedures, if implemented, may allow further positive transformation within the quality of spine care offered.

## Figures and Tables

**Figure 1 jpm-13-00614-f001:**
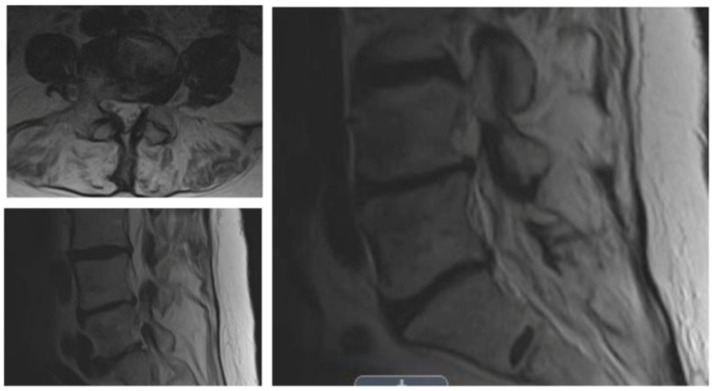
The patient vignette given to responding surgeons described a 66-year-old female with a 6-month history of progressive leg and back pain consistent with neurogenic claudication unresponsive to conservative treatments. The patient consistently rated her symptoms > 50/100 on the visual analog scale (VAS) for back pain, and her associated disability > 40% with the Oswestry Disability Index (ODI).

**Figure 2 jpm-13-00614-f002:**
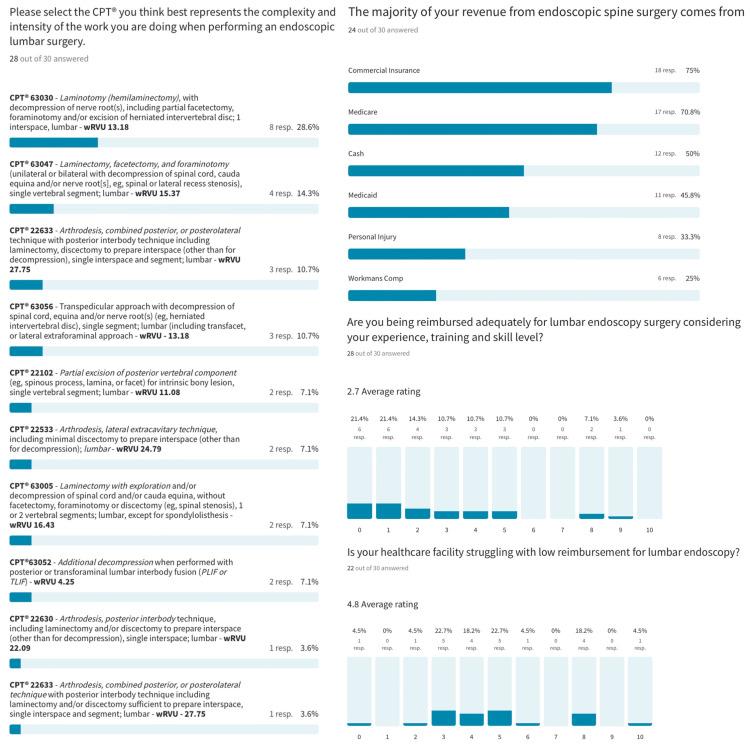
Of the responding surgeons, 85.8% valued the wRVU over 13, 46.6% over 15, and 14.3% over 20. Considering multiple responses, surgeons’ revenue for the endoscopic procedure was most frequently generated by commercial insurance (75%), Medicare (70.8%), cash payments (50%), Medicaid (45.8%), personal injury settlements (33.3%), and workmen’s compensation (25%). Only 3 of the 30 surgeons who filled out a complete survey thought they were adequately paid considering the complexity and the intensity of the lumbar endoscopic decompression surgery. Most surgeons (78.5%; <50th percentile) did not think they were adequately compensated. Regarding facility reimbursement, 77.3% of surgeons reported that their healthcare facility struggled to cover the cost with the received compensation.

**Figure 3 jpm-13-00614-f003:**
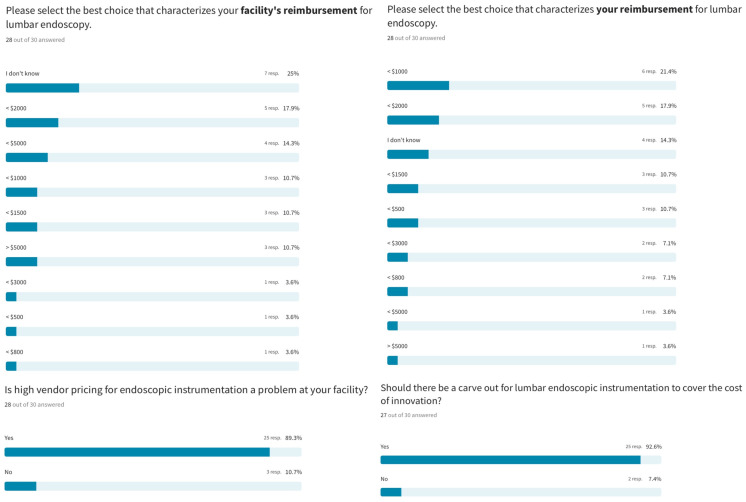
Regarding facility reimbursement for lumbar endoscopic spine surgery, 25% of responding surgeons did not know what their facility was paid. Four surgeons (14.3%) receiving cash payments indicated their facility’s payment was under USD 5000. The majority (46.5%) reported their facility receiving less than USD 2000, while 10.7% reported receiving less than USD 1500 and 17.9% reported receiving less than USD 1000. The professional fee received by surgeons was <USD 1000 for 21.4%, <USD 2000 for 17.9%, and <USD 1500 for 10.7%, meaning the fee was less than USD 2000 for 50% of responding surgeons. Only three surgeons (10%) were paid more than USD 2000 for lumbar endoscopic spine surgery. The majority (89.3%) reported high vendor pricing as problematic at their healthcare facility, making the case (92.6%) for an endoscopic instrumentation carveout to pay for the added cost of the technology.

**Figure 4 jpm-13-00614-f004:**
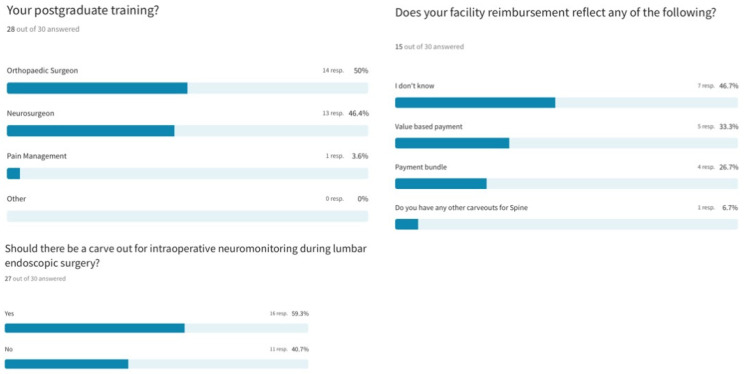
Most responding endoscopic spine surgeons had post-graduate subspecialty training in orthopedic surgery (50%), followed by neurosurgery (46.4%) and pain management (3.6%). Additional carveouts for intra-operative neuromonitoring during awake lumbar endoscopic surgery were recommended by 59.3% of responding surgeons. Regarding payment models, 46.7% of surgeons did not know what arrangement their healthcare facility had with the payors regarding reimbursement of endoscopic spine surgery. Value-based and bundled payment models were reported by 33.3% and 26.7% of surgeons, respectively.

**Table 1 jpm-13-00614-t001:** CPT^®^ codes representative of 2018 wRVUs cited from the Medicare fee schedule.

CPT	Non-Facility		Facility		
ADA HCPCS	Work RVU	PE RVU	PE RVU	MPE RVU	Description
22,212	20.99	17.68	17.68	5.73	Partial excision of posterior vertebral component (e.g., spinous process, lamina, or facet) for intrinsic bony lesion; single vertebral segment; lumbar
22,532	25.99	19.20	19.20	7.83	Arthrodesis; lateral extracavitary technique, including minimal discectomy to prepare interspace (other than for decompression); thoracic
22,533	24.79	18.10	18.10	5.96	Arthrodesis; lateral extracavitary technique, including minimal discectomy to prepare interspace (other than for decompression); lumbar
22,612	23.53	16.98	16.98	6.31	Arthrodesis; posterior or posterolateral technique; single level; lumbar (with lateral transverse technique when performed)
22,630	22.09	17.38	17.38	7.12	Arthrodesis; posterior interbody technique, including laminectomy and/or discectomy to prepare interspace (other than for decompression); single interspace; lumbar
22,633	27.75	18.89	18.89	7.96	Arthrodesis; combined posterior or posterolateral technique with posterior interbody technique including laminectomy and/or discectomy sufficient to prepare interspace (other than for decompression); single interspace and segment; lumbar
62,380	0.00	0.00	0.00	0.00	Endoscopic decompression of the spinal cord or nerve root(s), including laminotomy, partial facetectomy, foraminotomy, discectomy, and/or excision of herniated intervertebral disc; 1 interspace; lumbar
63,005	16.43	13.50	13.50	5.39	Laminectomy with exploration and/or decompression of spinal cord and/or cauda equina without facetectomy, foraminotomy, or discectomy (e.g., spinal stenosis); 1 or 2 vertebral segments; lumbar, except for spondylolisthesis
63,030	13.18	11.67	11.67	3.96	Laminotomy (hemilaminectomy) with decompression of nerve root(s), including partial facetectomy, foraminotomy, and/or excision of herniated intervertebral disc; 1 interspace; lumbar
63,047	15.37	12.73	12.73	4.53	Laminectomy, facetectomy, and foraminotomy (unilateral or bilateral with decompression of spinal cord, cauda equina, and/or nerve root(s), e.g., spinal or lateral recess stenosis); single vertebral segment; lumbar
63,056	21.86	15.52	15.52	6.63	Transpedicular approach with decompression of spinal cord, equina, and/or nerve root(s) (e.g., herniated intervertebral disc); single segment; lumbar (including transfacet or lateral extraforaminal approach) (e.g., far-lateral herniated intervertebral disc)
63,620	15.60	11.77	11.77	5.77	Stereotactic radiosurgery (particle beam, gamma ray, or linear accelerator); 1 spinal lesion

Source: OWCP Medical Fee Schedule, effective 30 June 2021 [23]. RVU—relative value units; PE—practice expense.

**Table 2 jpm-13-00614-t002:** Conversion factors (CFs) used in the last 7 years to calculate revenue generated by spine surgeons [25].

Year	Conversion Factor (CF)
2016	USD 35.8043
2017	USD 35.8887
2018	USD 35.9996
2019	USD 36.0391
2020	USD 36.0896
2021	USD 34.8931
2022	USD 34.6062

## Data Availability

The data presented in this study are available upon request from the corresponding author.

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
