# Peer review of "Determination of Work Related to Endoscopic Decompression of Lumbar Spinal Stenosis"

_jpm, 2023, doi:10.3390/jpm13040614_

Round 1

Reviewer 1 Report

This manuscript is an investigative study on whether the payment for endoscopic spinal surgery is reasonable. It is worth noting that the development of a new technology requires the timely updating of relevant payment standards, so as to ensure a virtuous circle. In this manuscript, the author conducted a detailed survey on the workload and operation fee of endoscopic surgery of spine from technical and physical effect, risk, intensity, and time during the pre-operative aspects through a questionnaire, which helps relevant departments to have a detailed understanding of the charging standard for diagnosis and treatment of such surgery. However, there are still some issues that need further discussion. However, there are still some issues that need further discussion, as follows:

1.This survey was only completed by 30 surgeons, and the sample size seems too small, which may affect the survey results.

2. The objectivity of the survey results may be affected by the fact that only surgeons complete the questionnaire survey without including staff in such industries as administration and insurance.

3. Lack of quantitative criteria for some evaluation indicators, such as operation time, average hospital stay, etc.

Author Response

Manuscript ID: jpm-2242325

Type: Article

Title: DETERMINATION OF WORK RELATED TO ENDOSCOPIC DECOMPRESSION OF LUMBAR SPINAL STENOSIS

Reviewer #1 Comments:

This manuscript is an investigative study on whether the payment for endoscopic spinal surgery is reasonable. It is worth noting that the development of a new technology requires the timely updating of relevant payment standards, so as to ensure a virtuous circle. In this manuscript, the author conducted a detailed survey on the workload and operation fee of endoscopic surgery of spine from technical and physical effect, risk, intensity, and time during the pre-operative aspects through a questionnaire, which helps relevant departments to have a detailed understanding of the charging standard for diagnosis and treatment of such surgery. However, there are still some issues that need further discussion. However, there are still some issues that need further discussion, as follows:

Response:       We thank this reviewer for recognizing that the payment cycle always lags behind the innovation.

Question #1:    This survey was only completed by 30 surgeons, and the sample size seems too small, which may affect the survey results.

Response:       This survey was aimed at US surgeons. Unlike in Europe or Asia, endoscopic spinal surgery is still not mainstream in the United States. Exceptionally few surgeons perform it, and even fewer have extensive experience with the procedure to render a meaningful opinion. Of note, 212 surgeons were reached, 82 started the survey, and 30 completed it. The response rate is typical for this type of survey. The 30 surgeons that did respond had extensive experience with endoscopic spine surgery. The final response number is likely a reflection of the dynamic explained above. We added a few sentences to that effect in the discussion and hope that reviewer #1 is satisfied with this explanation.

Question #2:    The objectivity of the survey results may be affected by the fact that only surgeons complete the questionnaire survey without including staff in such industries as administration and insurance.

Response:       We respectfully disagree with the reviewer on this point. The surgeons are the ones operating, and only they can assess the workload involved in the surgery. Perhaps the reviewer is eluding to low reimbursement for the industry and its support staff. If so, facility reimbursement needs to be addressed in a different study, as these payments are typically used to remunerate vendors. To solicit information from payors made no sense to us as they are currently undervaluing endoscopic spine surgery and have no inherent interest in paying more. The downward trend in reimbursement, contrary to the increasing healthcare cost across the board, is the problem and remains a primary hurdle to innovation and its clinical implementation. We hope reviewer #1 accepts our explanation.

Question #3:    The objectivity of the survey results may be affected by the fact that only surgeons Lack of quantitative criteria for some evaluation indicators, such as operation time, average hospital stay, etc.

Response:       We purposefully did not limit the assessment work involved in performing the endoscopic spine surgery to operative time or other perioperative measures of workload. Skill levels among surgeons may vary widely. One surgeon may be able to do the operation quickly, and another may take an excessive amount of time to accomplish the desired clinical outcome. Of note, endoscopic spine surgery in the United States is also primarily performed in an outpatient surgery center setting. Hence, admissions to the hospital are uncommon, and tracking such data would be of little benefit. Using the clinical vignette and comparing not just the operative time but also the presurgical and postsurgical time invested into the treatment episode, the intraoperative complexity, and intensity – the three commonly employed categories by the AMA in assigning an RVU value to a new code proposal – in the authors' mind is a much more comprehensive way of assessing workload and adequate payments. Reducing this process to operative time would likely result in an oversimplification and undervaluing of the work involved in endoscopic spine surgery. One aspect of endoscopic spine surgery may illustrate this dynamic. For example, in traditional spine surgery, the exposure and closure time required to create a surgical access corridor to the painful pathology, control bleeding, and close the wound may be substantial once the operation's goals have been accomplished. The AMA considers this "low intensity" time since no actual decompression or reconstructive work is done on the spine during these operation portions. In endoscopic spine surgery, creating access with the placement of the endoscopic working channel may take little time. Still, once the surgeon is docked at the surgical pathology, the intensity and complexity of tasks and problem-solving that must be done immediately without much downtime are instantly high. The entire operation is considered more stressful than open surgery because of the unfamiliarity with the videoendoscopically visualized pathology, the limited access, and the high-risk nature of complications. We hope the review sees our point and can accept our methodology of using a clinical vignette as a superior methodology.

Reviewer 2 Report

Manuscript ID: jpm-2242325

Type: Article

Title: DETERMINATION OF WORK RELATED TO ENDOSCOPIC DECOMPRESSION OF LUMBAR SPINAL STENOSIS

 Review Comments:

This study assesses the endoscopic procedure's technical and physical effort, risk, and overall intensity without focusing on the time required to perform the surgery. The presented data were received from 30 responding surgeons. The surgeons' responses were analyzed to determine comparator CPT codes and estimated wRVUs. This study is valuable, and the results could have a significant impact on spinal surgeries. The paper is original and well-written. A few comments may be considered to enhance this paper.

1.  The authors may present the statistical method in the article.

2. Are the responding surgeons enough? Was a power analysis done about whether the number of data was sufficient in the study?

Author Response

Manuscript ID: jpm-2242325

Type: Article

Title: DETERMINATION OF WORK RELATED TO ENDOSCOPIC DECOMPRESSION OF LUMBAR SPINAL STENOSIS

Reviewer #2 Comments:

This study assesses the endoscopic procedure's technical and physical effort, risk, and overall intensity without focusing on the time required to perform the surgery. The presented data were received from 30 responding surgeons. The surgeons' responses were analyzed to determine comparator CPT codes and estimated wRVUs. This study is valuable, and the results could have a significant impact on spinal surgeries. The paper is original and well-written. A few comments may be considered to enhance this paper.

Response:       Thank you.

Question #1     The authors may present the statistical method in the article.

Response:       The statistical methods are inherent to the typeform website which generates the descriptive statistics and graphical reports. We added a sentence to that effect in the method description.

Question: #2.  Are the responding surgeons enough? Was a power analysis done about whether the number of data was sufficient in the study?

Response:       The explanations to the reviewers comment were given in the response to a similar question by reviewer #1.

Round 2

Reviewer 1 Report

I am basically satisfied with the author's answers to several questions.